# A DIFFERENTIABLE BLEU LOSS.
# ANALYSIS AND FIRST RESULTS

**Noe Casas**[*†], **José A. R. Fonollosa**[*], **Marta R. Costa-jussà**[*]
[*] TALP Research Center. Universitat Politècnica de Catalunya. Barcelona, Spain.
[†] United Language Group.
contact@noecasas.com, {jose.fonollosa,marta.ruiz}@upc.edu

### ABSTRACT

In natural language generation tasks, like neural machine translation and image captioning, there is usually a mismatch between the optimized loss and the *de facto* evaluation criterion, namely token-level maximum likelihood and corpus-level BLEU score. This article tries to reduce this gap by defining differentiable computations of the BLEU and GLEU scores. We test this approach on simple tasks, obtaining valuable lessons on its potential applications but also its pitfalls, mainly that these loss functions push each token in the hypothesis sequence toward the average of the tokens in the reference, resulting in a poor training signal.

## 1 INTRODUCTION

Currently dominant sequence transduction architectures receive source tokens and target prefix tokens as input, and they output a categorical probability distribution over the target token space for the next position in the sequence. They are normally trained to minimize the categorical cross entropy between the generated probability distribution and the expected token one-hot representation or its *label smoothed* form (Szegedy et al., 2016). In some application fields like machine translation, model evaluation is driven by the BLEU score (Papineni et al., 2002), which computes a corpus-level measure of the amount of n-gram matches between reference and generated sequences. Despite the problems of the BLEU score (Doddington, 2002; Callison-Burch et al., 2006) and the availability of other automatic measures like METEOR (Banerjee & Lavie, 2005), BLEU is the *de facto* evaluation standard for machine translation research.

This mismatch between training and evaluation criteria has been addressed in the past by trying to incorporate BLEU scores as rewards in a REINFORCE (Williams, 1992) loss, like in (Ranzato et al., 2015). Using the same setup, scores other than BLEU have also been used, e.g. GLEU score (Wu et al., 2016). REINFORCE, however, suffers from the high variance of its gradient estimation. The need for such an estimation instead of directly having a BLEU loss derives from the dificulty of propagating gradients through the discrete stochastic units at the output of NMT models (recently proposed approaches include using directly the output of the softmax or using the Gumbel-softmax reparameterization (Jang et al., 2017; Maddison et al., 2017)), and from the fact that the BLEU score and the other available automatic measures are not differentiable, so they cannot be incorporated in the loss computation directly. Here we present a differentiable implementation of the BLEU and GLEU scores that tries to address the second problem, first describing the computations themselves in section 2 and then testing them under simple tasks in section 3. Our findings, however, suggest that BLEU and GLEU are poor training signals, as they lead the trained model to generate an averaged vector in token space, as described in section 5.

## 2 DIFFERENTIABLE BLEU AND GLEU SCORE LOSS FUNCTIONS

BLEU score is the *de facto* evaluation measure for machine translation tasks. GLEU score was proposed in (Wu et al., 2016) as a surrogate of BLEU. We propose a differentiable implementation of the computations of the BLEU and GLEU scores, taking as input a reference sequence $ref$ and a hypothesis sequence $hyp$ where each token in either of them ($ref_i$ or $hyp_i$) belongs to the simplex $\Delta_v = \{p \in \mathbb{R}^v : p_i \geq 0, \sum p_i = 1\}$, where $v$ is the size of the vocabulary, that is, they are

potentially smooth one-hot vector representations of the token. We assume both $ref$ and $hyp$ have the same maximum length. By means of mere algebraic operations, described in detail in appendix A, we devise a hard-wired (i.e. non-trainable) fully differentiable implementation of the BLEU and the GLEU scores at minibatch level. This enables us to define a loss function to directly optimize on the same criterion as the evaluation phase. Instead of directly optimizing $\mathcal{L} = -score$, we opt for the more numerically stable logarithm $\mathcal{L} = -\log(score(translation(source), ref) + \varepsilon)$, where $source$ is the source sequence, $translation$ represents a differentiable machine translation model and $\varepsilon$ is a small constant to ensure the logarithm receives values that are greater than zero.

## 3 EXPERIMENTS

### 3.1 CORRECTNESS VALIDATION

In order to validate the correctness of our BLEU and GLEU implementations, we compared their results with the ones from the NLTK implementation (without smoothing nor multibleu emulation). We confirmed that when strict one-hot vector inputs were used, the obtained results were precise at least up to $10^{-3}$ (with the scores expressed in $[0, 1]$). If the inputs are not strictly one-hot, like the output of a softmax, the results start diverging. The lower the maximum probability within the distribution, the larger the divergence.

### 3.2 TOKEN COPYING TASK

Here we replicate the token copying experiment from (Zhukov & Kretov, 2017). The task consists in copying a reference sequence of one-hot encoded tokens. The *model* is a sequence of logits of the same length as the reference, which are applied a softmax function to obtain the corresponding tokens. As loss, we make use of our BLEU and GLEU score implementations. In order to combine the output of the *model* with the score computation, we directly make use of the softmax of the logits. The obtained results, described in detail in appendix B, show that the BLEU and GLEU losses only obtain good results for very short sequence lengths but, when we increase the sequence length or vocabulary size, the obtained scores drop drastically. An analysis is provided in section 5.

### 3.3 SEQUENCE REVERSAL TASK

This experiment consists in a sequence reversal task using a sequence-to-sequence with attention model (Bahdanau et al., 2014; Luong et al., 2015) (see section 3.4 for justification of the model selection), using as loss function our differentiable BLEU and GLEU scores. The used data is the source language side of the WMT14 English-German set. The obtained results, which are described in appendix C, show that the losses are far from the scores obtained by the cross-entropy loss. An analysis of these results is provided in section 5.

### 3.4 MACHINE TRANSLATION TASK

In our early experiments, we used the Transformer architecture (Vaswani et al., 2017) as model for a translation task on the WMT 2014 English-German dataset. The outputs of the network were connected to our differentiable GLEU score by means of soft Gumbel-softmax sampling. However, the network quickly learned to abuse the GLEU loss: for each token, it always generated the previous gold data token. This way, the score in training was very good but in test time it was very poor. This is the analysis of the problem:

At training time, the decoder of the Transformer architecture receives as input the gold target sequence. By construction, the prediction generated by the model for a token cannot take advantage of the gold tokens neither at the position being generated or at the following positions, that is, each token's computation can only *see* the previous tokens. This is addressed by the masked self-attention mechanism in the decoding part. The traditional cross-entropy loss training tries to match each specific token and the network learns to generate it based on the prefix. However, when using the BLEU/GLEU scorer as loss, the network can directly profit from *seeing* the past gold data as the n-gram matches are independent from the position; this way, for each token to be predicted it can learn to generate any of the previous gold tokens (which are visible) in order to lower the loss, hence

learning to generate a sequence that is exactly the same as the expected one but shifted one position to the right (e.g. for an expected output of "the dog is blue", the system would learn to generate "xxx the dog is blue", where "xxx" is some other token generated by the network). This strategy gives perfect BLEU/GLEU scores in training, when the gold data prefix is visible. However, during inference, this kind of trained model generates the same token for all the positions. Therefore, using this kind of loss together with gold data knowledge (i.e. teacher forcing (Williams & Zipser, 1989)) implies a leakage problem. This led us to switch to models that work without receiving as input the gold data, namely sequence-to-sequence with attention (Bahdanau et al., 2014; Luong et al., 2015) trained with backpropagation through time and not teacher forcing. The results obtained were also poor, but the lesson learned about leakage avoidance was deemed worthy for inclusion in this report. Scheduled Sampling (Bengio et al., 2015) is another option that should be evaluated in the future.

## 4 RELATED WORK

Despite being devised independently, our work is similar to the ideas by Zhukov & Kretov (2017). They propose a differentiable lower bound of the BLEU score. For this, they compute n-gram matches matrices, like we do. The main aspects of their work that differentiate it from ours are: 1) the brevity penalty is not part of the differentiable path of the computation, 2) the repetition of n-grams in the hypothesis sequence is not properly taken into account by the matrix formulation of their BLEU score (see how in their paper the summation in (8) and those in Appendix A are not equivalent), as the precision of repeated tokens is added several times, leading to an incorrect clipped match count, 3) the experiments carried out, which consist on a token copying task, lack thorough exploration of different configurations, as only a sequence of 10 tokens is tested; our tests with the released source code reveal that the obtained BLEU scores decrease drastically when increasing the vocabulary size or the sequence length, much like in our proposed approach.

Note that our BLEU implementation is differentiable with respect to its inputs. There are in the literature other implementations of BLEU whose differentiability is not with respect to its inputs, like the work by Rosti et al. (2011) and hence we do not consider our work related to them.

## 5 ANALYSIS AND CONCLUSION

This paper presents differentiable implementations of the BLEU and GLEU scores and evaluates their performance on simple tasks, with poor results. In order to understand this, we propose a reinterpretation of n-gram match loss functions that leads to better understanding of their problems: n-gram matching losses can be understood as a multi-task learning setup (Caruana, 1998) where there is one subtask per each combination of hypothesis and reference n-gram, which pushes the hypothesis n-gram tokens to match the reference ones. These subtasks pursue mutually incompatible goals, as each hypothesis n-gram can only be actually close to a specific value, not different values. This results in a combined loss function that pushes each hypothesis token toward the average of the tokens in the reference, resulting in a poor training signal. This suggests the need of a coordination mechanism that leverages those individual sub-losses to drive the trained system to high total scores. This is further illustrated in appendix D.

This way, the main lessons learned during this work are 1) that loss functions that are computed at sequence level might lead the model to abuse the available knowledge about gold data where the past gold data tokens are available to the network (e.g. when using teacher forcing), and 2) that n-gram matching losses like BLEU or GLEU *per se* are not enough to provide a good training signal and that further research is needed to find a way to combine the individual sub-losses properly.

### ACKNOWLEDGMENTS

We would like to thank the United Language Group (ULG) and the Catalan Agency for Management of University and Research Grants (AGAUR) for the Industrial PhD Grant supporting this work. This work is also supported in part by the Spanish Ministerio de Economía y Competitividad, the European Regional Development Fund and the Agencia Estatal de Investigación, through the postdoctoral senior grant Ramón y Cajal, the contract TEC2015-69266-P (MINECO/FEDER,EU) and the contract PCIN-2017-079 (AEI/MINECO).

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

## APPENDIX A   DESCRIPTION OF THE BLEU/GLEU COMPUTATION

At sentence level, both BLEU and GLEU compute the number n-gram matches (for $n$ in $\{1, 2, 3, 4\}$) between reference and hypothesis (i.e. true positives count: $tp$) and, from it, both derive the precision as the ratio between true positives and total n-grams count in the hypothesis ($tpfp$). While BLEU adds a brevity penalty ($BP$) and combines the precision for each n-gram level geometrically, GLEU defines a recall measure as the ratio between true positives and total n-gram count in the reference ($tpfn$), and computes the final score as the minimum between precision and recall (an equivalent computation is to divide the true positives between the maximum of the total n-gram counts of reference and hypothesis). At corpus level, instead of averaging the sentence-level scores, both BLEU and GLEU combine first all numerators and denominators, and then the final aggregation is computed[1] , as shown in (1).

$$BLEU = BP \cdot e^{\sum_n w_n \log \frac{tp_n}{tpfp_n}} \qquad GLEU = \frac{\sum tp}{\sum \max(tpfp, tpfn)} \qquad (1)$$

Now we present the differentiable version of the above computations. As most steps are common to both BLEU and GLEU, we shall present them together. Let $ref$ be the reference sequence of tokens and let $hyp$ be the hypothesis sequence of tokens. Without loss of generality, here we assume that both reference and hypothesis have the same maximum length, $T$. Each token of reference ($ref_i$) and hypothesis ($hyp_i$) belongs to the simplex $\Delta_v = \{p \in \mathbb{R}^v : p_i \geq 0, \sum p_i = 1\}$, where $v$ is the size of the vocabulary, that is, each token is a smooth (or not) one-hot vector representations of the token.

We assume that there is a special token, the end-of-sequence token (usually depicted as `` or `eos`) that marks the end of the sequence before reaching length $T$. Beyond ``, the tokens up to the end of the sequence are to be ignored. Let $ref\_length\_mask$ be a vector of length $T$, with values close to $1$ in the positions of $ref$ where the end-of-sequence token has not yet appeared and values close to $0$ at its position and after; it can be computed as shown in (2), where $eos$ is the end-of-sequence token index within a token vector.

$$ref\_length\_mask_t = \prod_{i=0}^{t}(1 - ref_{i,eos}) \qquad (2)$$

Based on it, the length of the sequence $ref$ can be computed as:

$$ref\_length = \sum_t ref\_length\_mask_t \qquad (3)$$

Then, the total n-gram count in $ref$ can be computed as:

$$tpfn = \sum_{n=1}^{4} \max(ref\_length - n - 1, 0) \qquad (4)$$

An analogous procedure can be followed to compute $hyp\_length\_mask$, $hyp\_length$ and $tpfp$.

Let $onegram(a, mask\_a, b, mask\_b)$ be a function that receives sequences $a$ and $b$ and returns a $s \times s$ matrix that expresses the 1-gram matches between $a$ and $b$, being $mask\_a$ and $mask\_b$ the respective length masks, in the same style as $ref\_length\_mask$. Therefore, we can compute the result of the function at position $(i, j)$ as:

$$onegram_{i,j}(a, mask\_a, b, mask\_b) = \langle a_i \odot mask\_a_i, b_j \odot mask\_b_j \rangle \qquad (5)$$

,where $\langle \cdot, \cdot \rangle$ represents dot product and $\odot$ represents element-wise product (Hadamard product), $a_i$ represents the $i^{th}$ token vector in sequence $a$ and $mask\_a_i$ represents the $i^{th}$ element in vector $mask\_a$. Note that these operations can be implemented in terms of convolutions.

Let $ref\_onegram\_matches$ be the result of applying function $onegram$ to $ref$ with itself. Let $hyp\_onegram\_matches$ be the result of applying function $onegram$ to $hyp$ with itself. Let $cross\_onegram\_matches$ be the result of applying function $onegram$ to $ref$ and $hyp$. With these three matrices, we perform the computations illustrated in figure 1

---

[1]NLTK's implementation of BLEU and GLEU is a helpful reference: `http://www.nltk.org/`

The computation described in figure 1 comprises the following steps:

- Computing a validity mask for the reference to avoid counting repeated elements in upcoming operations. In order to obtain such a mask, we take only the upper triangular part of $ref\_onegram\_matches$, subtracting each element of the result from 1 and collapsing the elements by multiplying them along one of the axes. The same type of mask is computed for the hypothesis.

- Computing the count of each 1-gram appearance in the reference, masking out the repeated elements with the validity mask previously computed. The same type of count is computed for the hypothesis.

- Multiplying $cross\_onegram\_matches$ by the hypothesis counts and the reference mask, each one along its associated axis. This gives us the count of matches, using the counts of the hypothesis, having the repeated elements masked out.

- Multiplying $cross\_onegram\_matches$ by the reference counts and the hypothesis mask, each one along its associated axis. This gives us the count of matches, using the counts of the reference, having the repeated elements masked out.

- Taking the minimum of these two last count matrices, obtaining the true number of matches (minimum between the appearances in the reference and in the hypothesis).

- Adding up all elements from this last matrix, which gives us the number of 1-gram matches.

This procedure can be applied in the same way to 2-grams, 3-grams and 4-grams, after having obtained the cross n-gram match matrix, which can be computed as:

$$cross\_ngram\_matches_{i,j} = \prod diag(cross\_onegram\_matches_{i:i+n-1,j:i+n-1}) \tag{6}$$

, which has dimensions $T-n+1 \times T-n+1$. That is, for each element, we extract a square window of width $n \times n$ and compute the product of its diagonal elements. Note that this can be computed in terms of a convolution by taking the logarithm and using an identity matrix as kernel, and then exponentiating back.

After following the procedure for computing the number of n-gram matches for each $n$, we only need to add them up together to obtain the number of true positive matches $tp$.

$$GLEU = \frac{\sum tp}{\sum \max(tpfp, tpfn)} \tag{7}$$

The GLEU score is finally obtained by dividing $tp$ by the maximum of $tpfp$ and $tpfn$, which were obtained at the beginning.

For BLEU, the only missing step is computing the brevity penalty ($BP$), whose role is to discourage candidate translations that with too short length with respect to the reference translation, and which can be obtained directly from $ref\_length$ and $hyp\_length$ as:

$$BP = \begin{cases} 0 & hyp\_length = 0 \\ 1 & hyp\_length > ref\_length \\ e^{1-ref\_length/hyp\_length} & hyp\_length \leq ref\_length \end{cases} \tag{8}$$

Finally, the BLEU score can be computed with its defining expression:

$$BLEU = BP \cdot e^{\sum_n w_n \log \frac{tp_n}{tpfp_n}} \tag{9}$$

The differentiable BLEU and GLEU computations can be extended to compute their scores over a minibatch by computing each sentence components ($tp$, $tpfp$, $tpfn$) and then combining them together before the final division for GLEU and the final multiplication and exponentiation in BLEU.

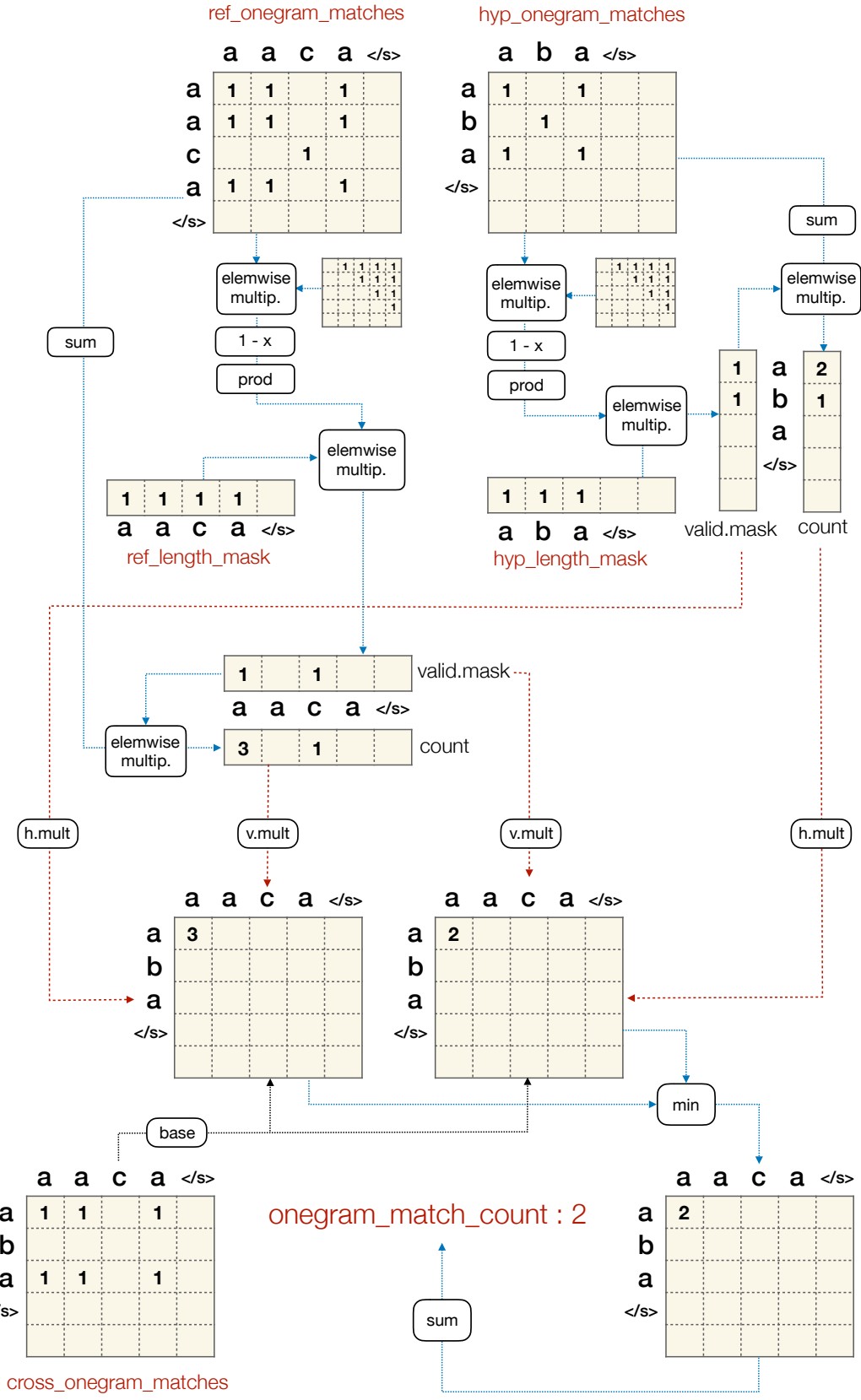

Figure 1: Example of 1-gram match computation, one-hot inputs.

## APPENDIX B    DETAILED RESULTS ON THE TOKEN COPYING TASK

This appendix provides detailed information of the results of the experiments described in section 3.2, which consist in a token copying task, where there is a reference sequence that is one-hot encoded and where the *model* is a logits matrix (with shape vocabulary size × sequence length). The goal is to make the logits match the reference tokens. The formulation of this task was taken from (Zhukov & Kretov, 2017).

We studied the behaviour of our BLEU and GLEU computations as training losses, connecting them directly to the softmax output (given the fixed nature of the reference and some preliminary test, we understood that Gumbel-softmax sampling hurts the performance of the training in this result and hence has not been further tested for this experiment).

The GLEU loss, whose results are shown in figure 2, is able to achieve 100 GLEU points in its copy only in the most simple configuration, with a sequence of length 10 and a vocabulary of 10000 tokens. As soon as any of these parameters are increased, the obtained scores decrease dramatically. It is not clear why for the simpler cases the performance is good and this is an aspect to be studied in the future.

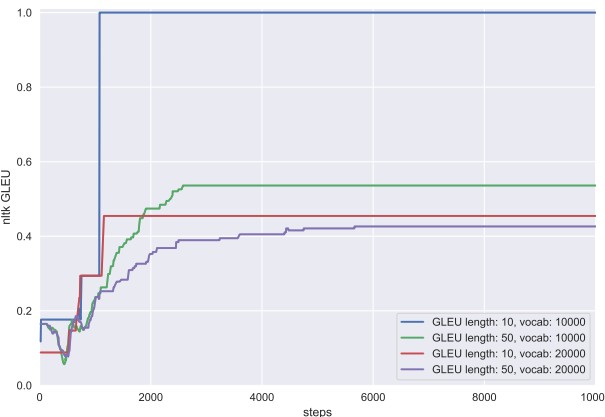

Figure 2: GLEU score during training of the model trained on the GLEU loss with different sequence lengths and vocabulary sizes.

The BLEU loss, whose results are shown in figure 3, obtains much worse results, not even reaching 40 BLEU in the best case.

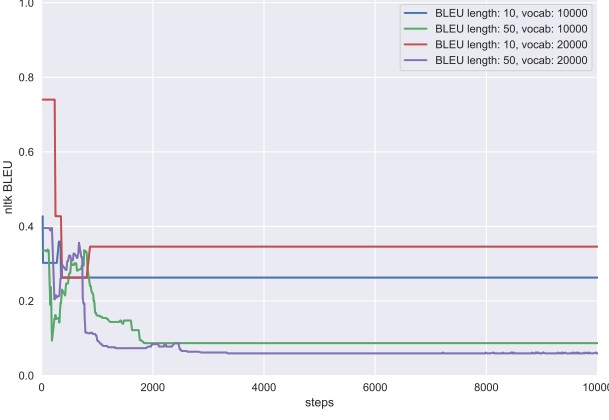

Figure 3: BLEU score during training of the model trained on the BLEU loss with different sequence lengths and vocabulary sizes.

## APPENDIX C   DETAILED RESULTS ON THE SEQUENCE REVERSAL TASK

This appendix provides detailed information of the results of the experiments described in section 3.3, which consist in a sequence reversal task. The data to be reversed is the source language side of the WMT 2014 training dataset. The target side is replaced by a reversed version of the source (not reversing the EOS and padding tokens). The trained model is a sequence-to-sequence with Bahdanau attention, with LSTM as units, with bidirectional encoder, embedding size of 256 and 256 units, maximum sequence length of 50 tokens, with a vocabulary of around 30K tokens. All the tests were performed with batch size of 40 and using Adam with learning rate of 0.001 during 10K iterations.

As described in section 3.4, the BLEU and GLEU losses cannot be applied to models with access to the gold data during training, so we train with back-propagation through time (BPTT) instead of teacher forcing. Nevertheless, as a performance reference for the experiments, figure 4 shows the performance of the model trained on this task with the normal cross-entropy loss, both with teacher forcing and BPTT. Note that teacher forcing achieves almost perfect scoring while BPTT only reaches 40 BLEU and its training is unstable. Note that the BLEU scores shown in the figure were computed with the NLTK implementation.

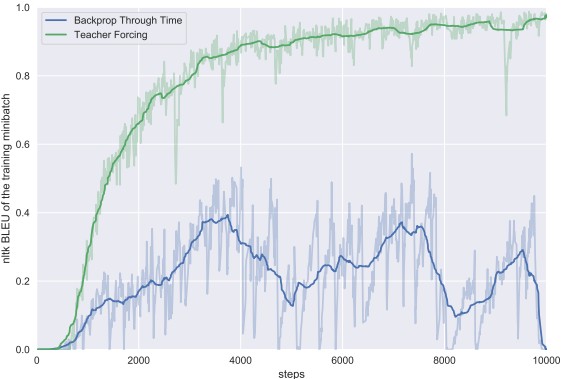

Figure 4: BLEU score during training of the model on BPTT and teacher forcing.

For the BLEU/GLEU-based training, we tested connecting the softmax directly to the score computation and using hard Gumbel-softmax sampling (G-S reparameterization with a straight-through estimator that turns the result into hard one-hot representations), with softmax temperature of 0.5. Figure 5 shows the poor resuls ontained by the GLEU score. Note that the GLEU scores shown in the figure were computed with the NLTK implementation.

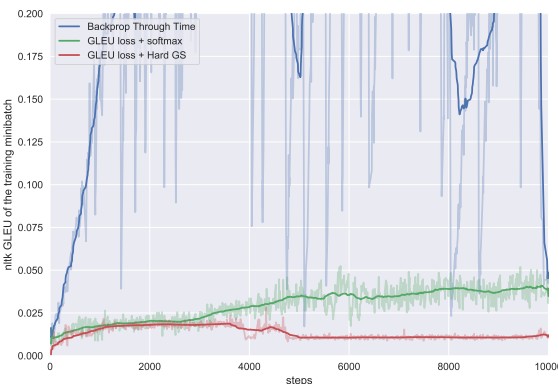

Figure 5: GLEU score during training of the model trained on BPTT, the GLEU loss fed with the softmax and the GLEU loss fed with hard Gumbel-softmax sampling. Y axis scale is [0, 0.2].

The training with the BLEU score had almost zero BLEU during all the training, so no figures were deemed necessary to illustrate its performance.

Note that the described training regime is similar to sampling a single sequence with REINFORCE. In the future we shall study how to leverage sampling in our approach to overcome high variance issues.

APPENDIX D    VISUALIZATION OF THE N-GRAM MATCH SUB-TASK LOSSES

The hypothesis presented in section 5 is that n-gram matching losses can be understood as a multi-task learning setup where each individual task consists in making each n-gram similar to each of the n-grams in the hypothes and that, given the symmetry of these mutually detrimental subtasks, the resulting combined loss function does not provide a good training signal as it pushes the weights towards the average of the reference sequence. Inspired by (Mescheder et al., 2017) and Ferenc Huszár's blog [2], we designed a version of the token copying task from section 3.2 that is simple enough to allow us to visualize the problem in two dimensions.

The simplified task consists in modeling a reference sequence of 2 bits ($ref$). For that, we will have a *model* with 2 real numbers as parameters; they are our hypothesis ($hyp$). They are applied a sigmoid to get numbers in the range $[0, 1]$. The loss function is defined as the cross entropy of the sigmoid of each component of the hypothesis against each component of the reference. This way, there are 4 sub-components of the total loss. This loss tries to mimic the way n-gram matches are aggregated in BLEU and GLEU. In figure 6 we show the vector fields of each of those sub-loss gradients together with the total loss gradient, using as reference sequence $ref = [0, 1]$; note that the cross-entropy is denoted as $J$. For each sub-loss we can see how the hypothesis component is pushed towards the corresponding reference component. However, when combining the individual losses, we get that the resulting gradient pushes the hypothesis toward the average of the bits in the reference sequence ($[0.5, 0.5]$).

This reasoning can be applied to the BLEU and GLEU losses, which follow the same principles in a higher-dimensionality parameter space. This way, our experiment suggests that our BLEU and GLEU losses need a further mechanism that combines the different pulling sub-losses into an actionable training signal. Some alternatives might be applying attention mechanisms, or following the line of assymetric multi-task learning approaches like (Kumar & Daume III, 2012) and (Lee et al., 2016). In future work we shall define further experiments that characterize the influence the other elements missing in the simplified bit copying experiment, like n-grams, using the geometric mean, imposing the brevity penalty, etc.

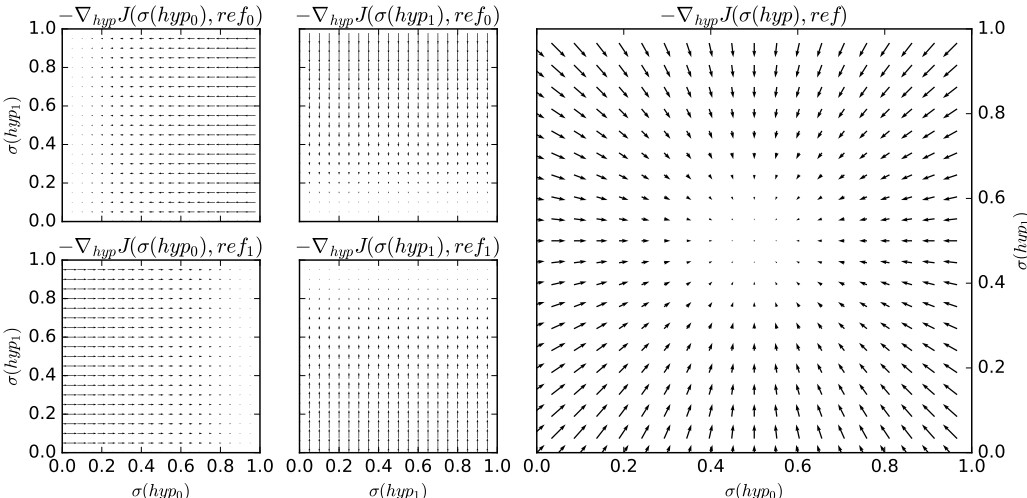

Figure 6: Vector field of the bit modeling losses for $ref = [0, 1]$. For each subplot, the x axis is $\sigma(hyp_0)$, the sigmoid applied to the first component of the hypothesis, and the y axis is $\sigma(hyp_1)$, the sigmoid applied to the second component of the hypothesis. Each of the small subplots represents the vector field of the gradient with respect to the hypothesis of the cross-entropy between one component of the hypothesis and one component of the reference, while the big subplot is the total loss or, equivalently, the addition of the vector fields from the small subplots.

---

[2] http://www.inference.vc/my-notes-on-the-numerics-of-gans/

