# OpenReview forum: "A differentiable BLEU loss. Analysis and first results"
_ICLR.cc/2018/Workshop — Accept_

### Official Review · AnonReviewer2 · 2018-03-04

**Rating:** 6
**Confidence:** 5

**Review:**

The paper presents a differentiable BLEU/GLEU loss, similar to Zhukov & Kretov (2017). This is an interesting research direction as it would make it easier to optimize the final automatic evaluation metrics our models are often judged by.

Related work: please also compare to earlier work on differentiable approximations to BLEU, e.g., Rosti et al. (2011), see below.

Section 3.4: Please describe the leakage problem more. Why does repeating seen gold tokens improve the score? Matching counts are still clipped.

In Appendix C, you say that you train with BPTT instead of teacher forcing. Teacher forcing inputs the reference token at every time step, which allows fast training since all time steps can be computed in parallel (for non-RNN architectures). Does this mean that you do not feed back the reference when you train with BPTT? In this case, your training regime seems similar to sampling a single sequence with REINFORCE. Your method would then also not be any more efficient than REINFORCE since sampling is by far the most expensive part of REINFORCE.

References:

Expected BLEU Training for Graphs: BBN System Description for WMT11 System Combination Task. Antti-Veikko I. Rosti, Bing Zhang, Spyros Matsoukas, and Richard Schwartz.

---

### Official Review · AnonReviewer1 · 2018-03-09
**Interesting analysis, methods section difficult to follow, some questions**

**Rating:** 7
**Confidence:** 4

**Review:**

This paper describes a differentiable version of BLEU, namely one that generalizes BLEU’s calculations involving 1-hot output sequences to sequences of distributions over the output vocabulary. It provides a number of experiments that lead to the conclusion that this idea does not work out-of-the-box, for reasons including incompatibility with teacher forcing, and a draw toward the mean token in the reference. This is a very dense 3-page paper that leans very heavily on its 6 pages of appendices, which explain the method and experiments in greater (but not perfect) detail.

Pros:

Interesting error analysis providing meaningful insight - the authors have demonstrated a desire to think deeply about this problem. Probably a good fit for a workshop paper.

Cons:

Unclear methods section.
Some questions linger regarding connection to previous work.
Some questions linger regarding the error analysis and resulting conclusions.

Clarity:

I didn’t like the methods section in Appendix A, which is presented almost entirely as a large bulleted list with narrative descriptions of mathematical operations. Surely this would have been easier and more precise with equations?

I am also unsure about some of the terminology in the paper. When confronted with a problem resulting from feeding gold-standard tokens into a position-independent n-gram matching objective, I think the authors opt to simply drop the previous input token from the NMT decoder state update. But I didn’t find their description to be particularly precise. It would be nice to specify any equation for the new network.

Quality & Significance:

This paper is hampered by its proximity to a NIPS 2017 workshop paper by Zhukov and Kretov on roughly the same topic. To their credit, the authors do a good job outlining some major differences between their differentiable BLEU formulation and their competitors’ in Section 4, emphasizing a modification to account for repeated n-gram in the hypothesis (I assume a connection here to BLEU’s clipping?), and accounting for brevity penalty. But I was still left with questions. The 2017 paper presents itself as a lower bound on expected BLEU to enable the REINFORCE algorithm without sampling, while this paper presents itself as a differentiable BLEU that provides an alternative to the REINFORCE algorithm, with no mention of bounds or expectations. Are these differences in presentation significant? I get the impression that they are, as this paper contrasts itself again the previous effort by saying, somewhat mysteriously, “their input is handled as a probability distribution, and their resulting score is also handled as such,” - How is this paper different? Surely by generalizing BLEU to the simplex, we are effectively treating the input as a probability distribution, or am I missing something? If I am missing a difference, what are the implications of the difference?

Furthermore, the 2017 paper uses Jensen’s inequality to create a bound on the BLEU’s product over n-grams - this paper doesn’t seem to do so, instead it appears to calculate BLEU directly with the soft matches. Has something been lost here? Or gained? It feels important. The paper should at least attempt to fill in these blanks for me.

On a similar note, I find myself very curious about the token copying experiment, which works well with sequences of length 10 and vocabulary sizes of 1,000, but poorly in all other scenarios. How is this related to the observation about the draw to the average token? Why does that particular configuration bypass this destructive pull toward the average, when all the other configurations fail? It is not intuitive to me at all why small vocabulary sizes would dodge this issue. Or is the copying experiment exhibiting another issue?

Do the experiments in Appendix D about the draw to the average token generalize when we consider the full BLEU score, that takes the geometric mean of 1-gram, 2-gram, 3-gram and 4-gram? While I can easily see the draw to the average in the unigram case, those same average tokens must be penalized very heavily in the bigram case, where to have a non-trivial probability of any one bigram, you would need confident scores on the simplex in adjacent positions for each of its component tokens. To support this hypothesis, if the draw to the average was that strong, you wouldn’t have seen the (technically correct) copying of the gold token at time t-1, when it was available.

Finally, despite citing the Concrete distribution and Gumbel-softmax in the introduction, the authors drop the input token altogether when confronted with the problem of the system mimicking the gold token provided from the previous timestep, instead of replacing the token with a distribution over tokens at the previous timestep as predicted by the current model, or with a sample from that distribution. Why is that? Is there any chance that by including the model-previous token in some form, we would see things improve?

Overall grade

I think this paper is a good fit for a workshop paper, mostly because it is not really an archival venue. Statements like those found in Section 5 can kill an approach for years, so I’m very nervous about letting such statements through while I still have so many questions about the approach and the analysis. But a workshop can help the authors hone their arguments much more rapidly, and hopefully lead us to better answers for a future archival version.

---

### Official Review · AnonReviewer3 · 2018-03-09
**Negative results on differentiable BLEU loss, but very interesting discussion.**

**Rating:** 6
**Confidence:** 3

**Review:**

This work proposes an differentiable BLEU loss which is expected to be used with REINFORCE for potentially better qualities measured by BLEU. Experiments are conducted on various tasks, and empirically shows that the results are mostly negative.

The proposed method look sounds, though it is hard to follow the details of the algorithm Appendix A. The negative results sound discouraging, but the discussion is interesting enough. I feel this work might spur discussion on investigating other trainable metrics for sequence problems.

---

### Decision · Program_Chairs · 2018-03-20
**ICLR 2018 Workshop Acceptance Decision**

**Decision:**

Accept

**Comment:**

Congratulations, your paper was accepted to the ICLR workshop.